# Development of an Evaluation Tool for Monitoring the Delivery of Psychosocial Care in Pediatric Oncology Settings

**DOI:** 10.3390/cancers17091550

**Published:** 2025-05-02

**Authors:** Kristin Foster, Bethany Sadler, Amy L. Conrad, Amanda Grafft

**Affiliations:** 1Department of Pediatrics, University of Iowa College of Nursing, Division of Hematology and Oncology, University of Iowa Stead Family Children’s Hospital, Iowa City, IA 52242, USA; 2Department of Pediatrics, Division of Hematology/Oncology, University of Iowa Stead Family Children’s Hospital, Iowa City, IA 52242, USA; bethany-sadler@uiowa.edu; 3Department of Pediatrics, Division of Pediatric Psychology, University of Iowa Carver College of Medicine, Iowa City, IA 52242, USA; amy-l-conrad@uiowa.edu (A.L.C.); amanda-grafft@uiowa.edu (A.G.)

**Keywords:** pediatric oncology, psychosocial oncology, program development, program evaluation, database, standards of care

## Abstract

Research indicates that the psychosocial and neurocognitive consequences of cancer and its treatment can have adverse lifelong effects, particularly when diagnosed and treated during childhood. After diagnosis, early psychosocial care is important for the timely identification of psychological and neurocognitive concerns and the initiation of appropriate interventions. A proactive approach involving assessment, psychosocial education, and intervention can reduce the intensity and severity of long-term psychosocial effects. The aim of our study was to use the 15 evidence-based standards of psychosocial care established by the Psychosocial Standards of Care Project for Childhood Cancer (PSCPCC) to develop a system for evaluating institutional progress in implementing these standards at the University of Iowa Stead Family Children’s Hospital (UI SFCH). This project resulted in the development of a REDCap^®^ version 10.4.1 electronic database, which allowed the pediatric oncology program to not only evaluate its current progress in implementing the PSCPCC standards at the time of this project but also continue to serve as a data repository for the longitudinal evaluation of the program. The initial data imported to the database included two cohorts of patients diagnosed with cancer in 2017–2018 (*n* = 68) and 2022–2023 (*n* = 82) for the comparison of psychosocial care delivery over time. The database is designed to track progress in each of the PSCPCC’s 15 standards of care. For the purpose of this article, data on Standard 1.A. will be shared to demonstrate the metrics produced by this tool and how they can be used to monitor the provision of services by the institution. With the ongoing use of this tool, the healthcare team at UI SFCH will be able to monitor program progress, identify gaps in psychosocial care, and evaluate specific interventions implemented to address these gaps.

## 1. Introduction

Despite significant increases in treatment effectiveness and marked improvement in long term survival rates, the diagnosis and treatment of cancer remains one of the most emotionally distressing events in medical care [1,2,3,4,5,6]. According to the National Comprehensive Cancer Network (NCCN), distress is defined as “a multifactorial unpleasant experience of a psychological (i.e., cognitive, behavioral, emotional), social, spiritual, and/or physical nature that may interfere with the ability to cope effectively with cancer, its physical symptoms and its treatment. Distress extends along a continuum, ranging from common normal feelings of vulnerability, sadness and fears to problems that can become disabling, such as depression, anxiety, panic, social isolation, and existential and spiritual crisis.” [4,5,6,7] fact, a large body of research documents the psychosocial risks for children and their families during and after cancer treatment, as well as approaches to reducing distress and supporting patients and families [2,8,9,10,11]. The Psychosocial Standards of Care Project for Childhood Cancer (PSCPCC; a group of pediatric oncology psychosocial professionals) collaborated with a larger interdisciplinary group of experts and stakeholders to develop evidence- and consensus-based standards for pediatric psychosocial care [10,12,13]. In 2015, this group published 15 evidence-based Standards for the Psychosocial Care of Children with Cancer and their Families (SPCCCF) [12].

The University of Iowa Stead Family Children’s Hospital (UI SFCH) pediatric oncology clinical care team has been dedicated to improving psychosocial care delivery in a manner that is consistent with these evidence-based standards. The clinical care team evaluated what resources were currently available and prioritized areas for improvement. Specifically, the team found that UI SFCH did not have a comprehensive team dedicated to the *routine and systematic assessment* of psychosocial healthcare needs and interventions, a situation reflective of national challenges in institutional psychosocial staffing and programmatic implementation [14,15,16,17]. The team responsible for caring for cancer patients included only one psychologist and one social worker, which was not enough staff to assess the number of patients in need or to follow the guidelines written within the standards. The team responsible for caring for cancer patients included only one psychologist and one social worker, which was insufficient to adequately assess the number of patients in need or to meet the standards outlined in guidelines. Although the team psychologist attempted to conduct an initial provider biopsychosocial assessment (PBA; focused on the holistic evaluation of biological, psychological, and social factors) for each patient in the early weeks following diagnosis, in reality, the demand for neuropsychological testing in other patients often prohibited the initial PBA of all new diagnoses. Even when an initial assessment was completed, unless a need for psychotherapy was identified at that time, follow-up evaluation did not typically occur due to limited provider resources [14,17,18]. In sum, the lack of a comprehensive psychosocial healthcare team resulted in reactive care as opposed to a proactive approach (taking initiative, preparing, and preventing future distress from occurring). A similar issue was found with respect to the social work assessments. While all newly diagnosed patients received at least one social work assessment (SWA; focused on social and environmental factors) shortly after diagnosis, there was no system in place for structured repeat assessments. Follow-up only occurred if needs were identified in the initial assessment or if the medical team later noted needs arising, typically in the case of an incipient crisis. As a result of its review, the clinical care team determined that an increase in psychosocial services was necessary and that the first step was to shift from a reactive to proactive model of care to align with SPCCCF Standard 1: “Youth with cancer and their family members should routinely receive systematic assessments of their psychosocial health care needs”. [12,18,19].

In January 2019, one full-time nurse practitioner was added to the existing clinical psychologist and medical social worker roles on the oncology team. The nurse practitioner was responsible for the PBA of all patients. A proactive model for these assessments was developed, with the goal of seeing all newly diagnosed patients within 4 weeks and at multiple time points throughout the cancer trajectory, including survivorship years [11,18,19,20,21]. The specific time points will be focused on in the Methods section. From these assessments, referrals to additional supportive services or interventions such as individual psychotherapy, neuropsychological testing, Child Life, or music therapy were made. The team’s clinical psychologist was able to focus her effort on neuropsychological testing and psychotherapy. Social work now had an added layer of support to turn to when patients and families were in need. Additionally, with the addition of the nurse practitioner, the oncology team now had a point person for prescribing and managing psychiatric medications [22,23].

Throughout the first two years of having a full-time nurse practitioner, there was a notable increase in the identification of psychosocial care needs in patients, resulting in further demand for intervention and resources. However, the team did not have a specific measurement and evaluation system in place to gather data that would objectively demonstrate the need for increased resources. The lack of such a system also prohibited measuring progress and identifying other areas for improvement in implementing the standards [18,24]. Ultimately, this prevented patients from receiving the levels of support set out in the SPCCCF standards.

Further guidance on implementation and evaluation was disseminated in 2020 by the psychosocial experts who had published the original SPCCCF standards of care [18,19,25]. In consultation with patient advocates, additional pediatric oncology experts and other stakeholders, they developed a matrix and guidelines to help healthcare providers identify and overcome barriers to the implementation of the SPCCCF standards of care. This was achieved through a rigorous, iterative review process including inputs by multidisciplinary psychosocial experts and focus groups and several rounds of revisions following additional expert reviews. This process resulted in the publication of a matrix and guidelines to help healthcare providers identify and overcome barriers to the implementation of the SPCCCF standards of care [18,19,25]. The matrix was designed to help clinicians assess the current levels of psychosocial care at their treatment sites, and the guidelines provide a variety of recommendations to help teams identify a pathway to achieving optimal psychosocial care. The matrix was designed with a scoring rubric with levels 1–5 aligned with each individual standard, with several broken down into sub- components of each standard. A score of 1 indicated a lack of implementation and a 5 indicated the complete implementation of the standards. Given the high level of variability in psychosocial resources across institutions, the matrix was designed with broad measurement criteria, leaving the flexibility for individual institutions to further define specific data points that fit within the broader definitions as they strive to evaluate their program successes and gaps.

To address the goal of developing a robust, comprehensive Pediatric Psychosocial Oncology Program in which a level of excellence in all 15 standards was achieved, the UI SFCH team utilized the rubric matrix and guidelines as tools to develop an institution-specific system for the evaluation of program progress in the implementation of psychosocial care delivery at UI SFCH. In this paper, we describe how objectives from the rubric matrix were framed and documented within a REDCap^®^ database (Research Electronic Data Capture, Vanderbilt University) [26,27,28], how data were extracted from medical records into this database, the current progress of our program, and the exploration of changes in effectiveness from prior cohorts. This system will continue to serve as a tool for the longitudinal evaluation of the program [24,29].

## 2. Materials and Methods

### 2.1. Human Subject Protection

Our prospective study was approved by the University of Iowa Institutional Review Board [IRB Form #202309467] with a partial waiver of HIPPA Authorization for all patients diagnosed prior to 7 November 2023. Children and adolescents who were diagnosed with pediatric cancer during the years 2017–2018 (n = 68) and 2022–2023 (n = 82) were recruited for our study, for a total of 150 patients. Figure 1 shows the process of recruitment for our study.

### 2.2. Procedures/Methods

This project began in 2017 (see Figure 2), when the first steps were taken to establish a psychosocial oncology program at UI SFCH. Using the 15 Standards for Psychosocial Care of Children with Cancer and their Families as the framework for overall program design, key gaps in care were identified. The current article focuses on Standard #1 (Systematic Assessment of Psychosocial Needs). This standard encompasses routine systematic assessments of both youth with cancer and their caregivers [6,7,30]. However, the metrics and data of focus in this article are specific to Standard 1.A., focusing on youth assessments. Gaps for this standard included (a) the timing and breadth of the initial assessment and (b) the frequency of follow-up assessments.

#### 2.2.1. Development of an Institution-Specific Matrix

The UI SFCH team modified the published matrix and guidelines [19] based on their existing capabilities to implement the standards (Table 1) and used the same Likert scale of 1–5 as the original matrix. The team began by scoring patients on Standard 1.A., which measures the team status on routine systematic assessments of youth with cancer throughout their cancer trajectory, specifically focusing on the provision of SWAs and PBAs.

#### 2.2.2. Chart Review Development

Using the Iowa-specific matrix as a guide, the team determined what datapoints available in the medical records would best provide information on how well the requirements within Standard 1.A. were addressed. These elements (Table 1) were extracted from the chart review and used as a foundation to structure a REDCap^®^ database.

#### 2.2.3. REDCap^®^ Database Collection

REDCap^®^ (Research Electronic Data Capture, Vanderbilt University) [25] is a secure and widely used online database system. Utilizing REDCap^®^ for data collection allowed the team to test draft designs and pick a design that fit the study and represented the UI data best [26]. Branching logic was used for yes or no questions and made data collection more efficient [27].

The REDCap^®^-based data repository has a total of 17 sections. The initial section is for basic demographic information related to the patients. The second section includes information regarding psychiatric diagnoses and identifies whether each psychiatric diagnosis occurred before or after the cancer diagnosis. Following the first two sections, the database includes individual sections dedicated to each of the 15 standards of psychosocial care. Each dedicated section is designed to collect important data to evaluate program performance with respect to the standard covered. The titles of each section within the database reflected the main points listed within each respective standard. Specific items from the chart review that were relevant to data collection for a given standard (e.g., the frequency of SWAs, the frequency of PBAs) were built into each dedicated section.

#### 2.2.4. Data Collection and Creation of a Training Manual

Data collectors used the EPIC^®^ healthcare records to extract patients’ information to answer questions within the section of the REDCap^®^ database pertaining to SPCCCF Standard 1 (see Table 1). To ensure that all collectors were retrieving information from the same place in EPIC^®^, a data collection manual was developed (available upon request from the corresponding author). This manual was used during collection and will be used to train future research assistants in data collection. Using the data import feature of REDCap^®^ allowed the team to download an Excel file from EPIC^®^ that could be reformatted for upload into the database. After the relevant data were transferred, the collectors would look at patients’ answers to questions and score them on the UI SFCH Matrix and Guidelines (Table 1).

### 2.3. Statistical Analysis

Scores from the UI SFCH Matrix and Guidelines were then exported from REDCap^®^ and imported to SPSS (Version 29.0) statistical software for all statistical analyses. To better understand the team’s functioning in relation to Standard 1.A., three one-tailed *t*-tests were used to determine whether there was a difference in the frequency of administration of the SWA and/or PBA between the two cohorts. The first looked only at the frequency of SWAs. The second looked only at the frequency of PBAs. The third looked at the combined score for the receipt of SWAs and PBAs. For each analysis, ninety-five percent confidence intervals were calculated, and significance was set at *p* < 0.05.

## 3. Results

### 3.1. Demographics

Two cohorts were included in this study, for a total of 150 patients. Table 2 summarizes the characteristics of each cohort. The participants’ mean age, averaged over both cohorts, was 8.93 (SD = 6.02), ranging from <1 year old to 25 years old. Most participants were male (56.67%, n = 85) and were white (82.67%, n = 124). In terms of disease state, most patients were part of the solid tumor group (38.67%, n = 58).

### 3.2. Standard 1.A. Matrix Scoring for Social Work Assessments

For each cohort, the social work assessments received by the patients were scored in accordance with the UI SFCH Matrix and Guidelines (Likert scale 1–5, where 5 is complete adherence) (Table 1). The mean social work score for the entire sample, that is, both cohorts, was 2.19 (SD = 0.880; 2.05–2.33, 95% Cl). This corresponds to patients receiving one social work assessment any time throughout their cancer trajectory. There was a significant difference between the two cohorts; the (2017–2018) cohort had a lower score on the matrix (mean = 1.0 [SD = 0.99]) compared to the (2022–2023) cohort (mean = 3.0 [SD = 0.93]; t(df) = 148.00, *p* < 0.01). Closer evaluation indicated that, while many patients received a social work assessment at some point during their treatment, it was rarely within one month of diagnosis. We did see improvement in this standard across two cohorts, where more patients were seen within one month of diagnosis in the 2022–2023 cohort. However, the mean score of this cohort was still at 3.0, indicating that timely follow-up assessment continues to represent a gap in care that needs to be addressed. Figure 3A shows the cohort scoring (Likert 1–5) for the UI SFCH Matrix and Guidelines.

### 3.3. Standard 1.A. Matrix Scoring for Provider Biopsychosocial Assessments

Patients were scored (Likert 1–5) with respect to their receiving PBAs according to the UI SFCH Matrix and Guidelines. The mean PBA score for the entire sample was 2.60 (SD = 2.27; 2.49–2.89, 95% Cl). There was a significant difference between the patients of the two cohorts, with those seen earlier (2017–2018) having a lower score (mean = 2.0 [SD1.24]) than those seen later (mean = 4.0 [SD = 1.20]; t(df) = 149.00, *p* < 0.01). Patients in the second cohort typically received an initial assessment, as well as at least one follow-up assessment. Figure 3B shows the scoring (Likert 1–5) for the UI SFCH Matrix and Guidelines.

### 3.4. Standard 1.A. Combined Matrix Scoring

Patients were scored on the combined score (Likert 1–5) with respect to their receipt of both assessments (SWA + PBA), as is written within the UI SFCH Matrix and Guidelines. The mean combined score for the entire sample was 2.10 (SD = 0.88; 2.05–2.33 95% Cl). There was a significant difference between the two cohorts, whereby those seen earlier had a lower score (mean = 2.0 [SD = 0.82]) than those who were seen later (mean = 3.0 [SD = 0.86]; t(df) = 148.00, *p* < 0.01). The second cohort of patients had a slightly higher mean score because they received both an initial SWA and one PBA. However, the timing of the SWA was delayed longer than was optimal (i.e., the 2022–2023 cohort patients were seen on average for the SWA over 1 month past their initial cancer diagnosis). Figure 3C shows the combined matrix scoring for both cohorts with the Likert scale.

## 4. Discussion

The present study aimed to evaluate the two patient care components of SWAs and PBAs in relation to the UI SFCH Matrix Guidelines. While the receipt of assessments did significantly improve between the two cohorts studied, data from all time points collectively support the hypothesis that Standard 1.A. is still not being fully implemented within our clinic. This highlights a specific gap in supportive care that needs further attention.

To the best of our knowledge, this is the first analysis exploring a way to create a data repository to perform evaluations of the implementation of the 15 evidence-based Standards of Care for Pediatric Oncology patients. As proof of principle, this approach successfully allowed the UI SFCH team to evaluate how well they were currently implementing Standard 1.A., that is, administering SWAs and PBAs as they are written within UI SFCH Matrix and Guidelines. As a result, the team was able to identify specific gaps in supportive care. A key aspect of the approach is the use of a database design (REDCap^®^) that facilitates longitudinal comparisons of cohorts. In summary, our analysis provides a practical demonstration of how, through the development of an electronic database, a clinical care team can adapt the Matrix and Guidelines included in the Pediatric Psychosocial Standard of Care Institutional Assessment Tool to perform an institute-specific evaluation of the implementation of the SPCCCF standards of care.

### 4.1. Receipt of Social Work Assessments

The qualitative results of our study identified key areas of importance to be used for evaluation and areas of improvement. The analysis of the receipt of SWAs showed that parents were only receiving one assessment any time throughout the cancer trajectory. Improvements were made for the later cohort in terms of a higher likelihood of a visit occurring within a month of diagnosis. However, the overall score for this standard remains low (mean = 3.0 for the most recent cohort), and ensuring that all parents of patients receive a visit within a month of diagnosis remains an unmet goal. To further improve implementation, the team should administer the first assessment at the time of diagnosis, have a follow-up assessment within 12 ± 3 months of the completion of treatment, and have annual assessments for the remainder of the individual’s lifetime. From a clinical perspective, if patients received more SWAs, this would allow the team to ensure that all the needs of the patients and caregivers were being accounted for, from the time of diagnosis, through the course and completion of the cancer treatment, and annually, over lifetime follow-up. For our clinic, a realistic next step will be ensuring that patients receive an assessment within one month of diagnosis and at least one additional follow-up.

### 4.2. Receipt of Provider Biopsychosocial Assessments

Given the increasing number of pediatric patients diagnosed with cancer who need supportive care *throughout* their cancer trajectory, the provision of PBAs is essential. This ensures that the patient’s need for support is identified early and interventions can be provided in a timely manner.

The analysis of the receipt of provider biopsychosocial assessment showed that patients in the earlier cohort were only receiving one assessment after the time of diagnosis. Patients seen later did have more evaluations, but their overall scores remained low due to the timing of evaluations—none were completed within one month of diagnosis. To improve the implementation of this standard, our team will need to reduce the time between diagnosis and the date of first PBA. From our clinical perspective, administering earlier initial assessment with frequent follow-up will help address needs proactively as they arise and may prevent a patient from experiencing a crisis.

### 4.3. Combined Matrix Scoring

The results of our current study, based on the combined score (SWA + PBA), suggest that, while at least one social work evaluation is being performed for all patients, the timing needs to improve. When evaluating Standard 1.A., achieving a score higher than 2 is not possible if the SWA is not completed within one month of diagnosis. With the PBA, one cannot obtain a higher score unless the provider(s) has completed an initial assessment at the time of diagnosis, during treatment, at the end of treatment, within the first year of completion, and annually over a lifetime. To improve standard-of-care implementation for future cohorts, the team’s initial goal would be to ensure that (a) patients were receiving an SWA within one month of diagnosis and (b) the PBAs are being administered at the time of diagnosis with at least one follow-up.


*Limitations*


Limitations in the study of these cohorts included a small sample size, a lack of patient demographic diversity, and the inability of the scoring system to capture all relevant services that had been received. The small sample size (n = 150) increases the risk of type II error (accepting a null hypothesis when a difference truly exists) and sampling bias. Our healthcare center serves a high proportion of patients from predominately White rural communities, this limited both the number and diversity of available patients. Healthcare centers that serve a more diverse population would have more patient representation in many different racial and ethnic groups. This may make the identification of other barriers (e.g., language barriers that may inhibit social work contact and the administration of assessments) more apparent. Another limitation was related to scoring constraints within the UI SFCH Matrix and Guidelines: many patients were reported as receiving multiple PBAs, but they were not administered within the time points recommended in the matrix, resulting in a lower overall PBA score. With SWAs, not receiving a follow-up SWA further prevented any PBAs from being represented in the combined scoring.


*Future Directions*


The findings of our evaluation study will be used to monitor the continuous improvement of the psychosocial oncology program here at UI SFCH. Specific attention will be focused on the identified areas for improvement, including the administration of SWA follow-up assessments and ensuring that the initial PBA is completed within a month of diagnosis-with regular follow-up. These data will be used as a baseline for future evaluative cohort comparison studies on Standard 1.A. While the team works on continuous improvement in relation to Standard 1.A., future research is needed to gain a deeper understanding of the impacts of social work and provider biopsychosocial assessments in relation to the caregivers of patients (Standard 1.B.).

## 5. Conclusions

The current research project demonstrated the lack of implementation in relation to Standard 1.A. for two cohorts of pediatric oncology patients and identified areas for improvement in terms of providing SWAs and PBAs for future patients. The monitoring of services was made possible using the electronic database REDCap^®^. The establishment of this database provided an organized way to compare cohorts and identify the gaps in care. Overall, this study showed the utility of implementation of a standardized system within a REDCap^®^ database project for evaluation of standard of care. This system will continue to allow the team to evaluate the success of new interventions and to identify areas of weakness that can be targeted for improvement to achieve the full implementation of the SPCCCF care standards.

## Figures and Tables

**Figure 1 cancers-17-01550-f001:**
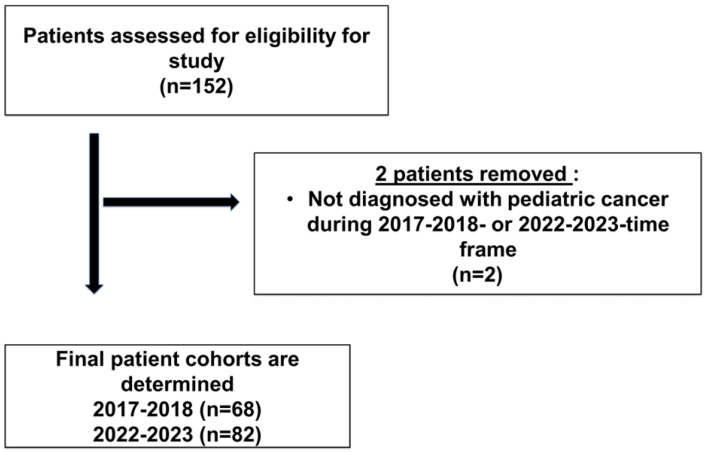
Flow chart of patient population/participant enrollment.

**Figure 2 cancers-17-01550-f002:**
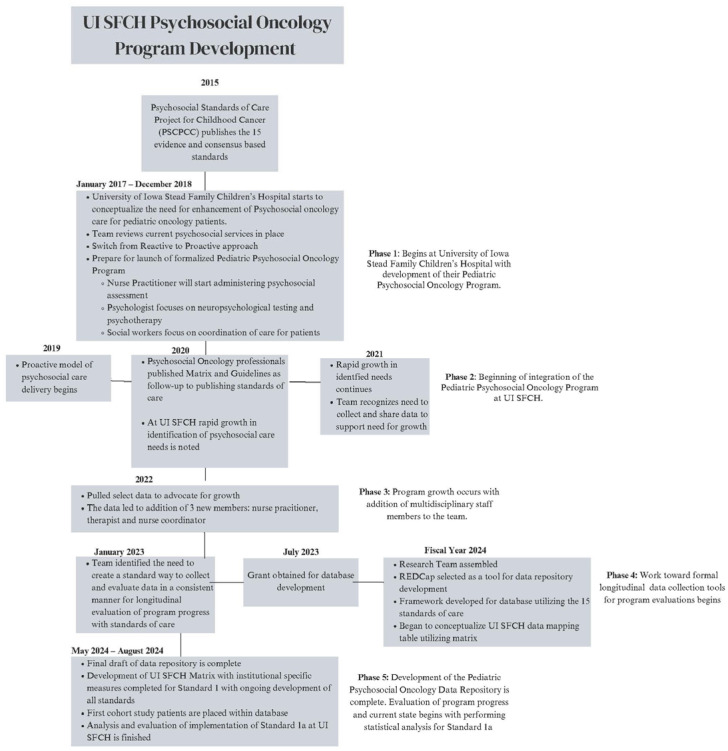
Development of Psychosocial Oncology Program at UI SFCH and development of program evaluation tool and data depository. The timeline includes the steps of development of the program, first cohort study, and first analysis using the evaluation tool that was designed by the team.

**Figure 3 cancers-17-01550-f003:**
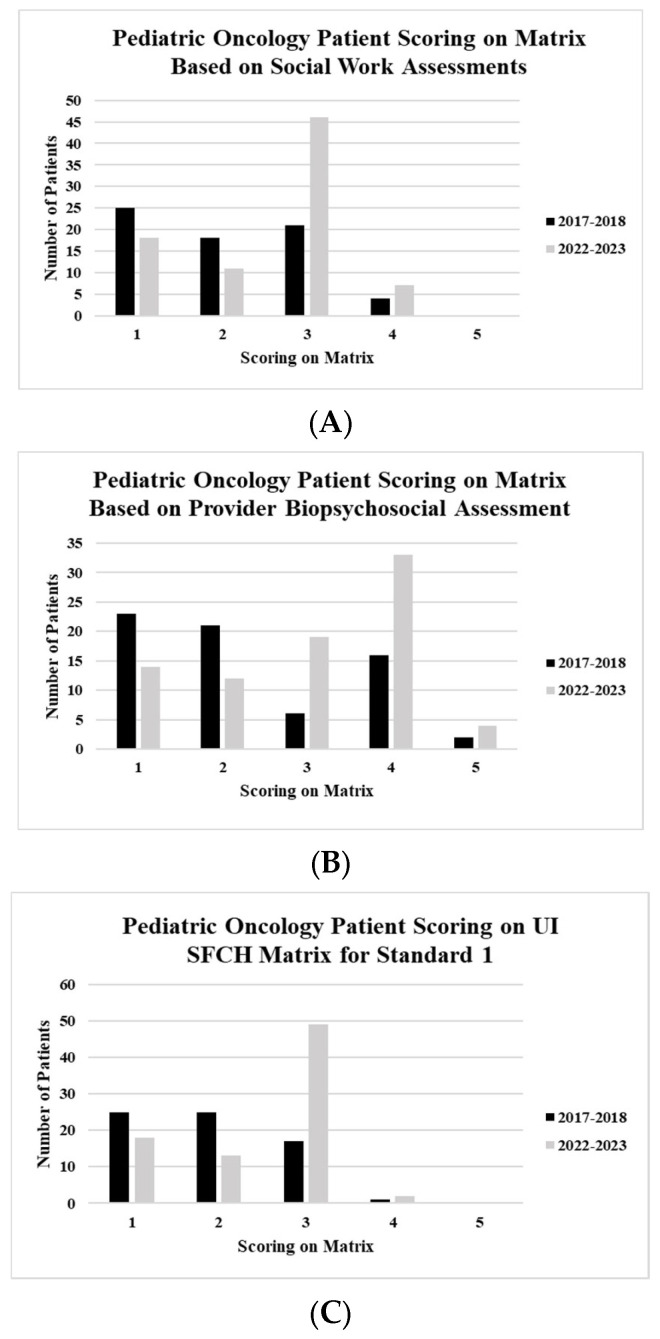
(**A**) The scoring of the patients within the cohorts based on receiving social work assessments according to the UI SFCH Matrix and Guidelines. (**B**) The scoring of the patients within the cohorts based on receiving provider biopsychosocial assessment according to the UI SFCH Matrix and Guidelines. (**C**) The scoring of patients within the cohorts based on receiving social work and provider biopsychosocial assessments according to the UI SFCH Matrix and Guidelines.

**Table 1 cancers-17-01550-t001:** University of Iowa Stead Family Children’s Hospital Matrix and Guidelines.

Standard 1. *Youth with Cancer and Their Families Routinely Receive Systematic Assessments of Their Psychosocial Healthcare Needs.*		
1: Consider that each of these have dimensions of the following: (a) periodicity (specified as at diagnosis, relapse/disease progression, and at end of treatment), (b) standardized process (systematic assessment), (c) content (see specified domains)		
1.A.: Assessment domains: Youth Youth pre-morbid and current adjustment Cognitive and academic functioning/concerns Developmental level and issues Family relationships Quality of social interactions Disease and treatment related concerns		
**Level**	**Original Matrix Scoring**	
1	No organized process in place for systematic assessments	
2	* *To be defined at an institution specific level*	
3	There is a system in place to assure that all youth receive assessment of psychosocial functioning early in the treatment trajectory and again only if clinically indicated	
4	* *To be defined at an institution specific level*	
5	All youth receives a comprehensive assessment at regularly scheduled points in their care	
**Level**	**UIHC Modified Scoring**	**Chart Review Elements**
1	No assessments completed on a child: Social Work Assessment (SWA) OR Provider Biopsychosocial Assessment (PBA)	Was a social work assessment completed? Y/N Date of social work assessment completion Dates of any repeated full SW assessments Frequency of SW check in during therapy Frequency of SW check in post therapy Psychosocial work assessment completed? Y/N Date of the initial assessment Dates of any follow-up assessment Time between visits (e.g., time in between first assessment and second assessment) Completed Oncology Treatment? Y/N (Date of completion) Psychosocial oncology assessment at the end of treatment? Y/N (Date of assessment)
2	Assessments completed as follows: Initial SWA completed at any time following diagnosis AND No PBA completed	
3	Assessments completed as follows: Initial SWA completed at diagnosis (within 1 month) AND One PBA completed any time after diagnosis	
4	Assessments completed as follows: SWA completed at least once at diagnosis (within 1 month) AND at least one SWA completed within the first year following the end of treatment (12 ± 3 months) AND PBA initial assessment completed AND at least one additional follow-up PBA completed	
5	Assessments completed as follows: SWA completed at least once at diagnosis (within 1 month) AND Annually (12 ± 3 months) following the end of treatment and continuing for a lifetime AND PBA at the following time points: Initial (within 4 weeks of diagnosis), minimum of every 3 months throughout treatment, end of treatment (±2 months of treatment completion), and at least twice within the year following treatment completion (±3 months), and Annually (12 ± 3 months), continuing for a lifetime	

**Table 2 cancers-17-01550-t002:** Demographics of patients.

	2017–2018 Cohort		2022–2023 Cohort	
	N	%	N	%
Age at Diagnosis				
<1 year old	3	4	9	11
1–2 years old	9	13	11	13
3–4 years old	11	16	8	10
5–7 years old	13	20	7	9
8–10 years old	7	10	12	15
11–14 years old	12	18	16	20
15–18 years old	9	13	19	22
19–25 years old	4	6	0	0
Diagnosis Group				
Non-oncology transplant	1	1	1	1
Leukemia/Lymphoma	22	32	20	24
Neuro-oncology	14	21	34	41
Solid tumor	31	46	27	34
Racial Identity				
White	54	79	70	85
Asian American	0	0	0	0
African American	5	7	4	5
Hispanic/Latino	5	7	6	7
American Indian/Alaskan Native	0	0	0	0
Native Hawaiian/Pacific Islander	0	0	0	0
Multiracial	4	6	2	3
Sex				
Female	24	35	41	50
Male	44	65	41	50
Intersex	0	0	0	0

## Data Availability

De-identified data available upon request and with approval from respective institutional human subject’s research review boards.

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
