# Peer review of "Development of an Evaluation Tool for Monitoring the Delivery of Psychosocial Care in Pediatric Oncology Settings"

_cancers, 2025, doi:10.3390/cancers17091550_

Round 1

Reviewer 1 Report (Previous Reviewer 2)

Comments and Suggestions for Authors

Round 1 comments:

The study touches on an important issue but the organization of the manuscript is problematic.

The intruduction should provide in-depth, up-to-date data and current standard information on the subject. Instead there is a long paragraph on changing ptactices, approaches and resources in the centre. That should be moved to methods or results.

Some of the Metohd section describes results of the study.

There is frequently lack of detail and specificity in the manuscript.

For example 'Multiple gaps in care existed' What gaps?

'A structure and workflow were designed with a proactive approach to psychosocial assessment as its central component.'

'team knew from the start that the goal would be to achieve excellence in psychosocial care'

etc.

There are results in Discussion section, while there is essentially no discussion of the data (as related to published data etc.)

The manuscript is satisfactory for publication now.

Round 1 reply:

Comments 1: The introduction should provide in-depth up-to-date data about current standard information on the subject. Instead, there is a long paragraph on changing practices, approaches and resources in the centre. That should be moved to methods or results. Some of the methods is desribes the results of the study. 

Response 1: Thank you so much for your feedback. The introduction in this manuscript provides background information that is important for the reader to understand. The project really focuses on the development of the database for use for program monitoring in accordance with standards of care. The description of the events at UI SFCH in the introduction are only those that led to identification of the need to create a matrix specific to our institution that would have a measurable Likert Scale items that align with our team's workflow and resources. We do briefly review this again in the first paragraph of the Procedures and Methods section (Pg. 5 Line 169).  Ultimately, the primary result of this project is an existing database that can provide program monitoring and evaluation over time but Standard 1A data is provided to demonstrate the type of evaluation that is possible using this database. With this in mind, we are leaving this section unchanged. 

Comment 2: There frequently lack of detail and specificity in the manuscript. 

Response 2:  Thank you for this feedback based on detail and specificity of the paper. After making the recommended changes we have added new information into the introduction section (Pg. 3). 

Comment 3: Multiple gaps in care are mentioned but what are the multiple gaps? 

Response 3:  The gaps in care are mentioned in (Pg. 3 Lines 80-98). The goal of the manuscript is to highlight the big picture gaps and then use the evaluation system to find in greater detail what gaps in supportive care exists. 

Comment 4: A proactive approach is mentioned using the psychosocial assessment as it component. 

Response 4: The workflow and the proactive approach are designed within (Figure 2 on Pg. 6). Where the team starts off with background information, how the team created their goal and how they were going to work towards achieving that goal. 

Comment 5: Team knew from the start that the goal of would be to achieve excellence in psychosocial care. 

Response 5: This statement identifies the goal of delivering the best psychosocial care possible to patients and families at UI SFCH by focusing on full implementation of the 15 standards of care. Following this statement, it is identified that it would be essential to create an evaluation tool. To achieve excellence in care, the team must score a 5 in all domains of the matrix, but the published matrix gives broad guidance on Likert scoring due to need for the tool to be adaptable to all institutions which will have different resources available therefore there is a need to develop institutional specific tools to objective measure achieving excellence. 

Comment 6: There are results within the discussion section, make this more of a discussion (related to published data)

Response 6: Thank you for this feedback. To our knowledge this is the publication that mentions a way to track gaps in care, specifically evaluating the implementation in the 15-standards of care. What we are trying to highlight in this paper is the development of this database and how it will work. The data on Standard 1a is meant to serve as an example of what this database can analyze. Therefore, we did not intend to look deeply into the comparison of these findings with other studies or institutions. That will occur in future manuscript as we approach a more thorough evaluation of Standard 1 as a whole and move toward publication on this information. 

Round 2 comments:

The manuscript is generally not changed, I have the same reservations. Main problem is lack of good division between method, results and discussion, and lack of good justification of the composition and flow of the paper.

Round 2 reply:

Our team would like to thank the editors and reviewers for their review of our manuscript. The feedback was very helpful to improve the structure, organization and readability of our paper. We have taken the opportunity to address all comments in this round 2 revision and feel the changes to the paper enhance the flow and understanding of our paper. Recommendations are organized by sections with the first reviewers from last time listed first followed by the round 2 comments. Changes within the manuscript are highlighted using red font color.

[Comment]: The manuscript generally is not changed, I have the same reservations. Main problem is lack of good division between methods, results, discussion, and lack of good justification of the composition and flow of the paper.

[Response]: Changes for round 1 and round 2 revisions have all been highlighted in red font throughout entirety of the paper. Methods, results, and discussion have all been re-written, simplified, and unneeded information was eliminated. To work on the flow of the paper the team took apart each individual paragraph. Through doing this we looked at our topic sentences (first two sentences of every paragraph) and ensured that they summarized the main points in every paragraph. All topic sentences that did not describe the main idea of the paragraph were eliminated and edited. All changes to the entirety of the paper to improve flow and ensure there are distinct methods, results and discussion have been implemented in red color.

Round 3 comments:

The manuscript is satisfactory for publication now.

Reviewer 2 Report (Previous Reviewer 3)

Comments and Suggestions for Authors

Round 1 comments:

The authors review their experience with implementation of a standardized psychosocial assessment tool in their practice.

Were there any barriers to assessment completion noted by staff that would impact the quality or rigor of the assessment?

The authors have created a system by which gaps are successfully identified, but a system for implementation of interventions is lacking. Please outline in the discussion the planned next steps. 

Since no patients identify as intersex, is it relevant to list this row?

Please add statistical significance metrics for Figure 3 comparisons across timepoints. Please consider refining the presentation of these figures. The text size is quite small.

Please cite/reference standards for psychosocial care. 

Were any assessments correlated between patient reported outcomes or compliance with survey frequency. 

Please provide a justification for comparison of the cohorts using a one-tailed vs. two tailed test. Was this due to the lack of significant differences in effect sizes or is this choice appropriately represented by the hypothesis.

Please provide further detail on the quality control and QA metrics for the RedCAP database.

Please detail the plans for and anticipated potential barriers associated with the eventual tracking of all 15 psychosocial care standards. Do the authors expect compliance with the larger inventory both in terms of completeness across all standards and across multiple timepoints?

There are several misspellings and formatting issues: please proofread the manuscript carefully moving forward. 

-Misspelling: instituinal, demostrate 

-Indent error page 1 - abstract.

-Please correct the table formatting for Table 2 (horizontal lines extend beyond margins for header rows only.

Round 1 reply:

Comment 1: Were there any barriers to assessment completion noted by staff that would impact the quality of rigor of the assessment? 

Response 1: This an excellent question. Since the current manuscript is focusing on the development of the database overall as a program monitoring tool, we did not explore the barriers to assessment completion for the purpose of this paper. As we evaluate Standard 1 in full and focus on this more deeply, we will certainly explore barriers. Thank you for pointing out this important item to address when we move toward publication for Standard 1 overall. 

Comment 2: The authors created a system by which gaps are successfully identified, but a system for implementation of interventions lacking. Please outline in discussion the planned next steps. 

Response 2: Thank you for this helpful suggestion. We have included further information in the Future Directions section (Pg. 14 Lines 382-396) on plans for future use of this data to review existing gaps and determine interventions to implement and then re-evaluate over time with additional cohort comparisons.  

Comment 3: Since no patient identifies as intersex, is this relevant to the list this row? 

Response 3: This was included so all cohort comparisons studies would have the same categories even when completing comparison studies in the future. 

Comment 4: Please add statistical significance metrics for Figure 3, comparison across timepoints. Please consider refining the figures the presentation of these figures. The text size is quite small. 

Response 4: The information for the statistical significance can be seen on (Pg.12 lines 291-304). 

Comment 5: Please cite/reference standards for psychosocial care. 

Response 5: The references and citation can be found within the references section (Pg. 15 Lines 447-534) specifically (citations 1,2,3,4,5). 

Comment 6: Were there any assessments correlated between patient reported outcome or compliances with survey frequency. 

Response 6: This will certainly be an important component of future work, but patient reported outcomes were not apart of this current project. The purpose of this project was the design of the program monitoring system for the 15 standards of care but a future project will look at these pieces more in depth by gathering reported outcomes and other measures on the assessments themselves. 

Comment 7: Please provide justification for comparison of cohorts using one-tail vs two-tail t-test. Was this due to lack of significant differences in effect size of this choice appropiate represented by hypothesis. 

Response 7: To provide justification for the usage of a one-tail vs two-tail t-test modifications were made into the methods section. Reasoning behind this was to see if there was an increase or decrease in the receipt of assessments among two cohorts of patients. Allowing us to determine if our hypothesis is directional, which would not be seen if we choose a two-tailed t-test. The two-tail t-test would not allow us to detect direction. Which is why one-tail t-test was chosen to allow us to detect improvement or weakness in implementation of the standards statistically. 

Comment 8: Please provide further detail on quality control and QA Metrics for REDCap database.

Response 8: To implement quality control, data collectors will review their entry 2-3 times before moving onto next section. This ensures that no data is being missed, and that information is being imported accurately. When downloading the data dictionary sheets and performing statistical analysis, analyst is told to perform calculation twice and take an average to promote accuracy in calculations. Also, through using the data dictionary, we can calculate the percentage of error, to determine how far off our measurements are if we must. Most questions are formatted in yes or no format, allowing for clear definitive answers to be achieved questions (Pg. 9 Lines 221-240).

Comment 9: Please detail the plans for and anticipated potential barriers with eventual tracking of all 15 psychosocial care standards. Do the authors expect compliance with larger inventory both in terms of completeness across all standards and multiple timepoints? 

Response 9: Thank you for this excellent question. Your question led to the addition of some information within our future direction section. (Pg. 13 line 394). The upkeep of the database longitudinally will certainly be a significant task so the workflow will need to be designed. Which the team has created a document for collectors every 6 months to update information. 

Comment 10: There are several misspellings and formatting issues: please proofread the manuscript moving forward. 

Response 10: Thank you, this has been corrected.

Comment 11: Misspelling: institutional, demonstrate. 

Response 11: Thank you this has been corrected. 

Comment 12: Indent error page 1: abstract.

Response 12: Thank you this has been corrected see (Pg. 2 line 30). 

Comment 13: Please correct table 2 horizontal lines extend beyond margins. 

Response 13: Thank you this has been corrected see (Pg. 10).

Round 2 comments:

The authors have adequately addressed all of my concerns.

This manuscript is a resubmission of an earlier submission. The following is a list of the peer review reports and author responses from that submission.

Round 1

Reviewer 1 Report

Comments and Suggestions for Authors

This was a clearly written paper, which was easy to follow and understand. 

I have some minor comments to help improve the presentation and comments relating to improving the content. 

  1. Requires a proof-read to remove some tracked changes (red) and correct some punctuation. See lines 227, 242 and 286 for examples.
  2. I would suggest realigning the content of Table 1 and/or removing bullet points so that the words do not appear to be jumping left to right. 
  3. Figure 2 does not appear to have a clear resolution and so seems a bit blurred.
  4. I would recommend giving each Figure 3a, 3b and 3c their own figure titles to improve clarity. 
  5. Is there a reason why Table 2 has uneven age groups? 
  6. The discussion would benefit from references to the current literature to link your findings to what is already known. How do your findings compare to other institutions/countries? 
  7. While there are 30 references in the reference list, only 7 appear as intext citations. 

Author Response

Comment 1: Minor comments to help improve presentation relating to improving content. Requires proof reading to remove some tracked changes and correct some punctuation errors. See lines 227,242 and 286 for examples. 

Response 1: We have modified the punctuation within lines (227,242 and 286) to provide more clarity. 

Comment 2: I would suggest realigning the content of Table 1 and/or removing bullet points so that the words do not appear to be jumping from left to right. 

Response 2: Thank you for this feedback the table on Page 6 lines 201-202 was modified. 

Comment 3: Figure 2 does not appear to have clear resolution and so seems a bit blurred. 

Response 3: Thank you so much for this feedback, Figure 2 has been edited and modified on page 6 lines 190. 

Comment 4: I would recommend giving Figure 3a, 3b,3c their own figure titles to improve clarity. 

Response 4: Thank you for this recommendation the addition of having separate figure titles can be seen now on Page 11 Lines 279-289. 

Comments 5: Is there a reason why Table 2 has uneven age groups? 

Response 5: The reason behind this, was creating age categories that would allow for more separation of data. When completing data collection, it was realized that we had large number of populations within certain age groups, leading to us have planned differences within age groups. While also wanting to create enough categories for future cohort comparison studies to prevent having unequal sample sizes within each category. 

Comment 6:  The discussion would benefit from references to the current literature to link your findings to what is already known. How do your findings compare to other institutions/ countries? 

Response 6: Thank you for this feedback to our current knowledge this is the only study that describes a database that allows health care teams to track the 15-evidence based standards. This has allowed our team to track strengths and weaknesses in implementing the 15-evidence standards. 

Comment 7: While there are 30 references in the reference list, only 7 appears as intext citations. 

Response 7: Some of our references were used to gain background information regarding our study but were not directly referenced within this paper. They were all important references that supported the development of the project overall. 

Reviewer 2 Report

Comments and Suggestions for Authors

The study touches on an important issue but the organization of the manuscript is problematic.

The intruduction should provide in-depth, up-to-date data and current standard information on the subject. Instead there is a long paragraph on changing ptactices, approaches and resources in the centre. That should be moved to methods or results.

Some of the Metohd section describes results of the study.

There is frequently lack of detail and specificity in the manuscript.

For example 'Multiple gaps in care existed' What gaps?

'A structure and workflow were designed with a proactive approach to psychosocial assessment as its central component.'

'team knew from the start that the goal would be to achieve excellence in psychosocial care'

etc.

There are results in Discussion section, while there is essentially no discussion of the data (as related to published data etc.)

Author Response

Comments 1: The introduction should provide in-depth up-to-date data about current standard information on the subject. Instead, there is a long paragraph on changing practices, approaches and resources in the centre. That should be moved to methods or results. Some of the methods is desribes the results of the study. 

Response 1: Thank you so much for your feedback. The introduction in this manuscript provides background information that is important for the reader to understand. The project really focuses on the development of the database for use for program monitoring in accordance with standards of care. The description of the events at UI SFCH in the introduction are only those that led to identification of the need to create a matrix specific to our institution that would have a measurable Likert Scale items that align with our team's workflow and resources. We do briefly review this again in the first paragraph of the Procedures and Methods section (Pg. 5 Line 169).  Ultimately, the primary result of this project is an existing database that can provide program monitoring and evaluation over time but Standard 1A data is provided to demonstrate the type of evaluation that is possible using this database. With this in mind, we are leaving this section unchanged. 

Comment 2: There frequently lack of detail and specificity in the manuscript. 

Response 2:  Thank you for this feedback based on detail and specificity of the paper. After making the recommended changes we have added new information into the introduction section (Pg. 3). 

Comment 3: Multiple gaps in care are mentioned but what are the multiple gaps? 

Response 3:  The gaps in care are mentioned in (Pg. 3 Lines 80-98). The goal of the manuscript is to highlight the big picture gaps and then use the evaluation system to find in greater detail what gaps in supportive care exists. 

Comment 4: A proactive approach is mentioned using the psychosocial assessment as it component. 

Response 4: The workflow and the proactive approach are designed within (Figure 2 on Pg. 6). Where the team starts off with background information, how the team created their goal and how they were going to work towards achieving that goal. 

Comment 5: Team knew from the start that the goal of would be to achieve excellence in psychosocial care. 

Response 5: This statement identifies the goal of delivering the best psychosocial care possible to patients and families at UI SFCH by focusing on full implementation of the 15 standards of care. Following this statement, it is identified that it would be essential to create an evaluation tool. To achieve excellence in care, the team must score a 5 in all domains of the matrix, but the published matrix gives broad guidance on Likert scoring due to need for the tool to be adaptable to all institutions which will have different resources available therefore there is a need to develop institutional specific tools to objective measure achieving excellence. 

Comment 6: There are results within the discussion section, make this more of a discussion (related to published data)

Response 6: Thank you for this feedback. To our knowledge this is the publication that mentions a way to track gaps in care, specifically evaluating the implementation in the 15-standards of care. What we are trying to highlight in this paper is the development of this database and how it will work. The data on Standard 1a is meant to serve as an example of what this database can analyze. Therefore, we did not intend to look deeply into the comparison of these findings with other studies or institutions. That will occur in future manuscript as we approach a more thorough evaluation of Standard 1 as a whole and move toward publication on this information. 

Reviewer 3 Report

Comments and Suggestions for Authors

The authors review their experience with implementation of a standardized psychosocial assessment tool in their practice.

Were there any barriers to assessment completion noted by staff that would impact the quality or rigor of the assessment?

The authors have created a system by which gaps are successfully identified, but a system for implementation of interventions is lacking. Please outline in the discussion the planned next steps. 

Since no patients identify as intersex, is it relevant to list this row?

Please add statistical significance metrics for Figure 3 comparisons across timepoints. Please consider refining the presentation of these figures. The text size is quite small.

Please cite/reference standards for psychosocial care. 

Were any assessments correlated between patient reported outcomes or compliance with survey frequency. 

Please provide a justification for comparison of the cohorts using a one-tailed vs. two tailed test. Was this due to the lack of significant differences in effect sizes or is this choice appropriately represented by the hypothesis.

Please provide further detail on the quality control and QA metrics for the RedCAP database.

Please detail the plans for and anticipated potential barriers associated with the eventual tracking of all 15 psychosocial care standards. Do the authors expect compliance with the larger inventory both in terms of completeness across all standards and across multiple timepoints?

There are several misspellings and formatting issues: please proofread the manuscript carefully moving forward. 

-Misspelling: instituinal, demostrate 

-Indent error page 1 - abstract.

-Please correct the table formatting for Table 2 (horizontal lines extend beyond margins for header rows only.

Author Response

Comment 1: Were there any barriers to assessment completion noted by staff that would impact the quality of rigor of the assessment? 

Response 1: This an excellent question. Since the current manuscript is focusing on the development of the database overall as a program monitoring tool, we did not explore the barriers to assessment completion for the purpose of this paper. As we evaluate Standard 1 in full and focus on this more deeply, we will certainly explore barriers. Thank you for pointing out this important item to address when we move toward publication for Standard 1 overall. 

Comment 2: The authors created a system by which gaps are successfully identified, but a system for implementation of interventions lacking. Please outline in discussion the planned next steps. 

Response 2: Thank you for this helpful suggestion. We have included further information in the Future Directions section (Pg. 14 Lines 382-396) on plans for future use of this data to review existing gaps and determine interventions to implement and then re-evaluate over time with additional cohort comparisons.  

Comment 3: Since no patient identifies as intersex, is this relevant to the list this row? 

Response 3: This was included so all cohort comparisons studies would have the same categories even when completing comparison studies in the future. 

Comment 4: Please add statistical significance metrics for Figure 3, comparison across timepoints. Please consider refining the figures the presentation of these figures. The text size is quite small. 

Response 4: The information for the statistical significance can be seen on (Pg.12 lines 291-304). 

Comment 5: Please cite/reference standards for psychosocial care. 

Response 5: The references and citation can be found within the references section (Pg. 15 Lines 447-534) specifically (citations 1,2,3,4,5). 

Comment 6: Were there any assessments correlated between patient reported outcome or compliances with survey frequency. 

Response 6: This will certainly be an important component of future work, but patient reported outcomes were not apart of this current project. The purpose of this project was the design of the program monitoring system for the 15 standards of care but a future project will look at these pieces more in depth by gathering reported outcomes and other measures on the assessments themselves. 

Comment 7: Please provide justification for comparison of cohorts using one-tail vs two-tail t-test. Was this due to lack of significant differences in effect size of this choice appropiate represented by hypothesis. 

Response 7: To provide justification for the usage of a one-tail vs two-tail t-test modifications were made into the methods section. Reasoning behind this was to see if there was an increase or decrease in the receipt of assessments among two cohorts of patients. Allowing us to determine if our hypothesis is directional, which would not be seen if we choose a two-tailed t-test. The two-tail t-test would not allow us to detect direction. Which is why one-tail t-test was chosen to allow us to detect improvement or weakness in implementation of the standards statistically. 

Comment 8: Please provide further detail on quality control and QA Metrics for REDCap database.

Response 8: To implement quality control, data collectors will review their entry 2-3 times before moving onto next section. This ensures that no data is being missed, and that information is being imported accurately. When downloading the data dictionary sheets and performing statistical analysis, analyst is told to perform calculation twice and take an average to promote accuracy in calculations. Also, through using the data dictionary, we can calculate the percentage of error, to determine how far off our measurements are if we must. Most questions are formatted in yes or no format, allowing for clear definitive answers to be achieved questions (Pg. 9 Lines 221-240).

Comment 9: Please detail the plans for and anticipated potential barriers with eventual tracking of all 15 psychosocial care standards. Do the authors expect compliance with larger inventory both in terms of completeness across all standards and multiple timepoints? 

Response 9: Thank you for this excellent question. Your question led to the addition of some information within our future direction section. (Pg. 13 line 394). The upkeep of the database longitudinally will certainly be a significant task so the workflow will need to be designed. Which the team has created a document for collectors every 6 months to update information. 

Comment 10: There are several misspellings and formatting issues: please proofread the manuscript moving forward. 

Response 10: Thank you, this has been corrected.

Comment 11: Misspelling: institutional, demonstrate. 

Response 11: Thank you this has been corrected. 

Comment 12: Indent error page 1: abstract.

Response 12: Thank you this has been corrected see (Pg. 2 line 30)

Comment 13: Please correct table 2 horizontal lines extend beyond margins. 

Response 13: Thank you this has been corrected see (Pg. 10).

Reviewer 4 Report

Comments and Suggestions for Authors

The University of Iowa Stead Family Children's Hospital (UI SFCH) has formalized its Pediatric Psychosocial Oncology Program, implementing 15 evidence-based Standards for Psychosocial Care for Children with Cancer and Families. The program sought to enhance progress and tackle care gaps through objective measures. Staffing and cost barriers were addressed, and a REDCap® database project was launched to assess the program's efficiency. The REDCap® data dictionary facilitated statistical analysis, enabling efficient evaluation of the program's adherence to care standards. This database will identify areas for improvement and ensure comprehensive care for pediatric oncology patients and their families. Prior to the publication decision, certain matters need to be clarified.

  1. In Discussion section, the study found that 45% of patients scored a 2 on the social work assessment matrix, indicating poor adherence to the standard. The UI SFCH matrix requires an initial social work assessment within one month of diagnosis for a 3 score, and at least one additional assessment within 15 months of therapy for a 4 score. Although there was improvement in the 2022-23 cohort, the mean score remains at 3.0, indicating a gap in care that needs to be addressed. What do the findings indicate about patient adherence to the social work assessment matrix, particularly regarding the timing of assessments in relation to diagnosis and therapy?
  2. In Conclusion section, the study found that social work and provider biopsychosocial assessments received lower attention in the 2017-2018 cohort and increased in the 2022-2023 cohort. The electronic database, REDCap®, was used for monitoring services, allowing for comparisons and identifying gaps in care. The study demonstrated the utility of implementing a standardized system within REDCap® for evaluating standard care and identifying areas for improvement. The 2-year CIR showed no statistically significant. How did the implementation of a standardized system within the REDCap® database improve the evaluation of biopsychosocial assessments and monitoring of services from the 2017-2018 cohort to the 2022-2023 cohort?
Comments on the Quality of English Language

The English can be enhanced for clearer expression of the research.

Author Response

Comment 1: In Discussion section the study found that 45% of patients scored a 2, on social work matrix, indicating poor adherence to the standard. The UI SFCH Matrix requires an initial social work assessment within one month of diagnosis for a 3 score, and at least one additional assessment within 15 months of therapy for a 4 score. Although there is improvement in the 2022-2023 cohort, the mean score remains at 3.0, indicating a gap in care that needs to be addressed. What do the findings indicate about patient's adherence to the social work matrix, particularly timing of the assessment relation to diagnosis and therapy? 

Response 1: There is certainly still a gap in care present. This manuscript is focusing on the development of the database as a tool that will be capable of tracking improvements and continue gaps. The data shared is meant to demonstrate this ability, not necessarily focuses on specifics of what the findings indicate. One of the next steps will be to fully evaluate Standard 1 and examine these types of questions. 

Comment 2: In conclusion section the study found that social work and provider biopsychosocial assessment received lower attention in 2017-2018 and increased in 2022-2023 cohort. The electronic database, REDCap was used for monitoring services allowing for comparison and identifying gaps in care. The study demonstrated the utility of implementing a standardized system within REDCap for evaluating standard of care and identifying areas of improvement. The 2-year CIR showed not statistically significant. How did implementation of a standardized system within REDCap database improve the evaluation of the provider biopsychosocial assessments and monitoring of services from the 2017-2018 to the 2022-2023 cohort. 

Response 2: The creation of a database was not what led to the improvement being seen in biopsychosocial assessments between these two cohorts. The database was not created until after the addition of another nurse practitioner to the psychosocial team. It was this intervention and the need to formally track the impact of this and future interventions that led to creation of the database. However, moving forward the database will guide the team in knowing what standard of care have notable gaps which will help identification of necessary interventions. The database will then help track progress made following each intervention implemented. 

Round 2

Reviewer 2 Report

Comments and Suggestions for Authors

The manuscript is generally not changed, I have the same reservations. Main problem is lack of good division between method, results and discussion, and lack of good justification of the composition and flow of the paper.

Author Response

Our team would like to thank the editors and reviewers for their review of our manuscript. The feedback was very helpful to improve the structure, organization and readability of our paper. We have taken the opportunity to address all comments in this round 2 revision and feel the changes to the paper enhance the flow and understanding of our paper. Recommendations are organized by sections with the first reviewers from last time listed first followed by the round 2 comments. Changes within the manuscript are highlighted using red font color.

[Comment]: The manuscript generally is not changed, I have the same reservations. Main problem is lack of good division between methods, results, discussion, and lack of good justification of the composition and flow of the paper.

[Response]: Changes for round 1 and round 2 revisions have all been highlighted in red font throughout entirety of the paper. Methods, results, and discussion have all been re-written, simplified, and unneeded information was eliminated. To work on the flow of the paper the team took apart each individual paragraph. Through doing this we looked at our topic sentences (first two sentences of every paragraph) and ensured that they summarized the main points in every paragraph. All topic sentences that did not describe the main idea of the paragraph were eliminated and edited. All changes to the entirety of the paper to improve flow and ensure there are distinct methods, results and discussion have been implemented in red color.